# RLTime: Reinforcement Learning-Based Feature Attribution for Interpretable Time Series Models

## Abstract

Deep time-series models are widely used in healthcare and finance, where interpretability is essential. Explaining these models is challenging due to temporal dependencies, nonadditive feature interactions, and high-dimensional inputs. Recent approaches learn continuous masks under sparsity constraints to generate attribution maps. While effective, this method has two key limitations: it explores the combinatorial space of feature subsets myopically, often missing synergistic features, and suffers from a soft-to-hard gap, where soft masks used during training misalign with the discrete selections needed at inference. We introduce RLTime, a framework that learns discrete attributions through sequential information acquisition. A masked reconstruction network recovers the latent representation of the reference model from partially observed inputs, such that the *change in the reconstructed latent* after revealing a feature can be leveraged to quantify its marginal value. This signal defines rewards for a distributional reinforcement learning agent that iteratively unmasks features, balancing exploration and exploitation while operating directly in the discrete action space. The agent's value function scores the utility of revealing each feature, enabling a clear ranking of features and a non-myopic acquisition policy. Experiments on synthetic and real-world datasets demonstrate that RLTime significantly improves attribution quality, exploration-exploitation balancing, and interpretability.

## 1 Introduction

Deep learning models for time series data have made significant progress across a range of applications, including forecasting (Lim et al., 2021; Liu & Wang, 2024), anomaly detection (Hundman et al., 2018), and healthcare analytics (Miotto et al., 2016; Choi et al., 2016). These models excel at capturing complex temporal dependencies, leading to improvements in predictive performance. However, their inherent complexity often makes them function as black boxes, where understanding the reasoning behind specific predictions poses a difficult challenge (Doshi-Velez & Kim, 2017; Guidotti et al., 2018). This is a major drawback in high-stakes applications such as medical diagnosis (Miotto et al., 2016; Choi et al., 2016; Kaushik et al., 2020) or financial decision-making (Singh & Srivastava, 2017; Zhang et al., 2017; Shah et al., 2019) in which providing accurate predictions is in reality only half of the task. There is a growing demand for explainability methods that offer insight into the decision-making process of the model (Ribeiro et al., 2016; Liu et al., 2024a; Queen et al., 2023). This paper aims to address this need by introducing a novel framework for explainability in deep time series modeling, enabling trust and accountability without sacrificing predictive power.

A prominent approach for explainability in time series involves learning *sparse masks* over input features or channels to attribute importance. These discrete attribution masks are designed to select certain key features that drive model predictions, effectively identifying which aspects of the input contribute the most. However, optimizing binary masks is non-differentiable, and so prominent lines of work (Queen et al., 2023; Liu et al., 2024a) have leveraged *surrogate gradients* like Gumbel–Softmax (Jang et al., 2017) and straight-through estimators (STE) (Bengio et al., 2013) to train *soft* masks that are later hardened to discrete subsets. This strategy enables end-to-end training under a sparsity budget and has set strong baselines.

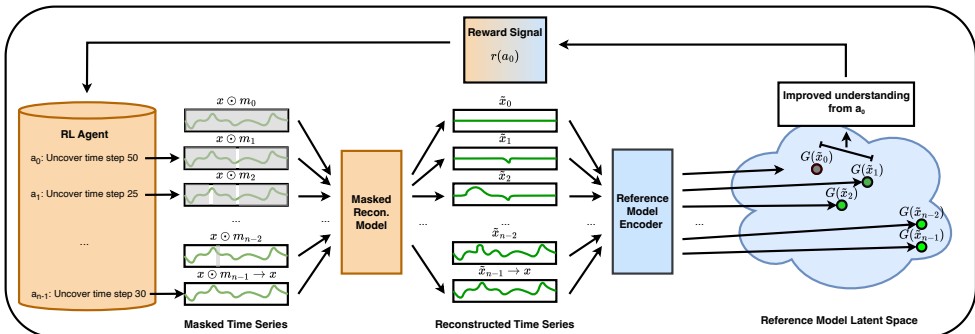

Figure 1: The RLTime training pipeline. The agent starts with a fully masked state. The agent sequentially selects features to unmask; unmasked features improve agreement with the reference model's latent (blue). The agent is trained on a *latent-shift* reward (squared change between consecutive reconstructed latents), and its learned values yield feature importance.

However, continuous relaxations are a brittle proxy for the discrete subsets that are actually evaluated. First, using a Gumbel-Softmax relaxation to select discrete features biases training toward **myopic exploration**, where features that yield immediate gradient reductions are reinforced while synergistic alternatives are prematurely pruned. As the distribution sharpens, this dynamic locks the model into suboptimal local minima, misaligning the soft relaxation with the discrete subset ultimately required. Second, a **soft-to-hard gap** arises through *fractional leakage*: during training, the relaxed mask disperses small weights across many complementary features and harvests their joint signal, but at evaluation, the mask is hardened to a sparse top-$K$, so this dispersion collapses, causing *interaction breakage* and misranking of features. In practice, these effects lead to poor exploration of the subset space and missed attributions, which ultimately degrade performance under discrete evaluation. Empiricial evidence are provided in Appendix C.

**Our approach.** We introduce RLTIME, a framework that recasts attribution as sequential information acquisition and operates natively in the discrete action space. We first pretrain a masked reconstruction network that, given a partially observed input, predicts the latent representation of a reference model. The *local latent shift*—the squared change in the reconstructed latent when a feature is revealed—quantifies its marginal value and becomes the reward. Under squared loss and a calibrated reconstructor, this latent shift equals the *expected* drop in latent MSE, yielding a principled, path-aware proxy for information gain. Using this reward, a distributional reinforcement learning agent, C51 (Bellemare et al., 2017), learns a value function over reveal actions with action masking to exclude already selected features. We use C51 instead of a scalar-value DQN because it learns a full return distribution on a fixed support, which captures how each feature's usefulness changes over time. A single value function suffices to both rank features at the initial state and to act step by step. Sorting expected returns at the start yields a transparent top-$K$ subset. Rolling the values forward produces a nonmyopic acquisition policy that naturally prices prerequisites, synergies, and heterogeneous costs. Training and evaluation both use the same discrete reveal actions, so RLTIME avoids the soft-to-hard mismatch of relaxed masks. This alignment removes fractional leakage, prevents interaction breakage at hardening, and yields attributions that better reflect the model's discrete dependencies.

We evaluate RLTime across a broad range of both real-world and simulated time series datasets, comparing it against a comprehensive set of recent baselines. Our results demonstrate significant and consistent performance improvements, establishing RLTime as the new state-of-the-art. Specifically, we observe a 63.52% average increase in Area Under Recall curve metrics, showcasing RLTime's superior ability to generate globally accurate and comprehensive feature attributions. Furthermore, RLTime's attribution maps are not only more accurate but also naturally smoother and more interpretable compared to prior methods, providing insights that are both actionable and reliable.

## 2 PROBLEM FORMULATION

We aim to develop an explanation framework for time series classification models operating over a time series dataset $\mathcal{T} = \{(x_i, y_i) \mid i = 1, \cdots, N\}$, where $x_i$ represents the input samples and $y_i$ denotes the labels for each sample. Each time series input $x_i \in \mathcal{X} \triangleq \mathbb{R}^{T \times V}$ is a matrix where $T$ represents the length of the time series and $V$ represents the number of variables. A *feature* here is defined as a time-variable pair, e.g. $x_i[t, v]$ being a feature representing the value of variable $v$ at time step $t$. The corresponding label $y_i \in \mathcal{C} \triangleq \{1, \cdots, C\}$ belongs to one of $C$ possible classes. A classifier model consists of two components: an encoder $G$ and a predictor $F$. The encoder $G$ encodes the input $x_i$ to a latent representation $z_i$, i.e., $G(x_i) = z_i \in \mathcal{Z} \triangleq \mathbb{R}^l$. Meanwhile, the predictor $F$ makes predictions based on the latent representation $z_i$, i.e., $F(z_i) = \hat{y}_i \in \tilde{\mathcal{C}} \triangleq [0, 1]^C$ where $\arg \max_j \hat{y}_j \in \mathcal{C}$ being the predicted label. Here we refer to $F \circ G$ as the reference model where $G$ is the reference encoder and $F$ is the reference predictor.

Feature attribution can be defined as a continuous map of the features that represents the relative importance of each feature to the prediction. For each input sample $x_i$, an attribution map $m_i \in \mathbb{R}^{T \times V}$ is given. For any time step $t_1$, $t_2$ and variables $v_1$, $v_2$, $m_i[t_1, v_1] > m_i[t_2, v_2]$ implies that $x_i[t_1, v_1]$ is a more important feature compared to $x_i[t_2, v_2]$.

## 3 METHODOLOGY

Our goal is to produce attribution masks that highlight which time–variable pairs $(t, v)$ most influence a trained reference model $F \circ G$ on $x \in \mathbb{R}^{T \times V}$. The central challenge is to score the *marginal value* of revealing a feature under partial observation, and to do so in a way that reflects the reference model's own internal representation rather than raw reconstruction fidelity alone. We address this in two stages. First, we pretrain a *reference-aware* masked reconstruction network that is consistent with the data manifold, the reference encoder's latent space, and the reference predictor's outputs. This yields a stable, model-faithful discrepancy we can measure on masked inputs. Second, we use the *stepwise reduction* of that discrepancy when a feature is revealed as a reward signal in an RL environment that searches the combinatorial space of masks.

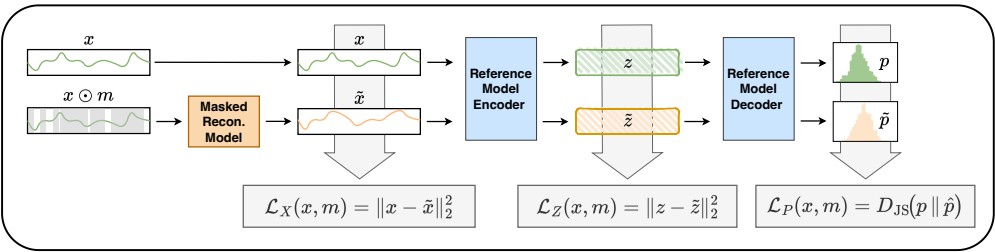

Figure 2: Diagram outlining our masked reconstruction training pipeline. Here, the input $x$ and masked input $x \odot m$ are aligned through three losses in three spaces: input, latent, and logit, respectively.

### 3.1 RECONSTRUCTION WITH CONSISTENCY

Under a mask $m \in \{0, 1\}^{T \times V}$, the reference model only sees a partial view of $x$, yet its latent representation $G(x)$ and prediction $F \circ G(x)$ are defined on the full input. To evaluate how informative a newly revealed feature is, we need a surrogate that (i) stays faithful to the data manifold, (ii) recovers the representation the reference model actually uses, and (iii) preserves its downstream prediction when appropriate. Plain input MSE alone is too sensitive to nuisance variation; matching logits alone can be flat and under-informative. We therefore combine complementary signals.

For each training example $(x, y)$, sample a binary mask $m \sim \mathcal{D}_{\text{mask}}$ and form a partially observed input

$$\hat{x}_m = m \odot x + (1 - m) \odot b,$$

where $b \in \mathbb{R}^{T \times V}$ is a fixed baseline, such as zeros or per-feature means. Here for all imputations, zero is adopted. A reconstruction network $R_\psi$ maps $\hat{x}_m$ to a reconstructed signal $\tilde{x} = R_\psi(\hat{x}_m)$. We pass both $x$ and $\tilde{x}$ through the frozen reference model:

$$z = G(x), \quad p = F(z), \quad \tilde{z} = G(\tilde{x}), \quad \tilde{p} = F(\tilde{z}).$$

We train $R_\psi$ with three losses that encourage consistency in input, latent, and prediction spaces:

$$
\begin{aligned}
\mathcal{L}_X(x, m) &= \|x - \tilde{x}\|_2^2 \quad \text{(input anchor)}, \\
\mathcal{L}_Z(x, m) &= \|z - \tilde{z}\|_2^2 \quad \text{(latent faithfulness)}, \\
\mathcal{L}_P(x, m) &= D_{\text{JS}}(p \,\|\, \tilde{p}) \quad \text{(prediction consistency)}.
\end{aligned}
\tag{1}
$$

The total objective can be written as:

$$\mathcal{L}_{\text{recon}} = \lambda_X \, \mathcal{L}_X + \lambda_Z \, \mathcal{L}_Z + \lambda_P \, \mathcal{L}_P, \quad \lambda_X, \lambda_Z, \lambda_P \geq 0.$$

$\mathcal{L}_X$ keeps $\tilde{x}$ near the observed signal so training remains stable under heavy masking and does not drift off-manifold. $\mathcal{L}_Z$ is the primary alignment signal: it teaches $R_\psi$ to reconstruct the *internal representation* $z = G(x)$ from partial inputs, which is exactly what we will measure to score marginal value. $R_\psi$ is trained with squared loss to predict $z$ from masked inputs, so its output $\tilde{z}$ approximates the *conditional expectation* $\mathbb{E}[z \mid I]$ given the currently revealed information $I$. We exploit this calibration in Section 3.3 to relate our stepwise latent-shift reward to expected reductions in latent mean-squared error. $\mathcal{L}_P$ tightens alignment when multiple inputs map to similar latents, encouraging reconstructions that preserve the prediction-relevant structure of $F \circ G$. This training regime is highlighted in Figure 2.

## 3.2 Feature Importance Metric

After training the reconstruction model, its reconstruction error in the latent representation space, $\mathcal{L}_Z(x, m)$, under a given mask $m$, can be used to assess the importance of the unmasked features. Specifically, if we compare two masks, $m_i$ and $m_j$, and observe that $\mathcal{L}_Z(x, m_i)$ is significantly lower than $\mathcal{L}_Z(x, m_j)$, this indicates that the unmasked features in $m_i$ allow for a much better recovery of the latent representation than those in $m_j$. Consequently, the features unmasked by $m_i$ are more critical to the reference model's prediction process. This relationship can be expressed as:

$$\mathcal{L}_Z(x, m_i) < \mathcal{L}_Z(x, m_j) \Rightarrow m_i \text{ is more important than } m_j. \tag{2}$$

We conduct an experiment on our synthetic datasets to validate this claim empirically, and highlight the results in Figure 3. For each dataset, we randomly select $K$ features, unmask only these features, and compute $\mathcal{L}_Z$. Here, $K$ corresponds to the known number of important features based on the dataset labels. Next, we iteratively replace the randomly selected features with the ground truth important features (as identified in the dataset) and recompute $\mathcal{L}_Z$ at each step.

Figure 3 demonstrates how $\mathcal{L}_Z$ evolves as the proportion of unmasked features transitions from random to ground truth important features. We observe a consistent and rapid decrease in $\mathcal{L}_Z$ as more important features are unmasked. This behavior supports the hypothesis that a lower $\mathcal{L}_Z$ corresponds to a better set of unmasked features. Furthermore, as the unmasked features approach the complete set of important features, $\mathcal{L}_Z$ approaches zero. This indicates that the remaining unimportant features contribute minimally to the composition of the latent embedding, confirming their irrelevance to the reference model's understanding.

**Relation to our reward.** While $\mathcal{L}_Z$ is a useful *set-level* diagnostic, our RL agent optimizes a *local* latent-shift reward between consecutive reconstructions. Under squared loss and a calibrated reconstructor, the *expected* latent shift equals the *expected* drop in $\mathcal{L}_Z$, so these criteria are aligned in expectation.

## 3.3 RLTime

**Environment.** We formulate the feature attribution task as an RL problem by framing it as a Markov Decision Process (MDP). An MDP is the 5-tuple $(S, A, R, p, \gamma)$ with state space $S$, action

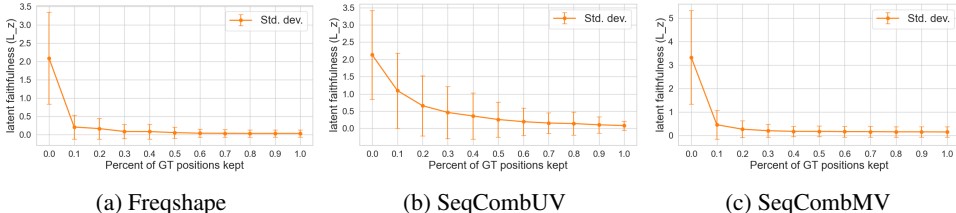

(a) Freqshape        (b) SeqCombUV        (c) SeqCombMV

Figure 3: Visualization of latent faithfulness $\mathcal{L}_Z$ as the percentage of ground-truth important features increases (random $\rightarrow$ ground truth). Lower $\mathcal{L}_Z$ is better, meaning the reconstructed latent is closer to the reference latent. Datasets: FreqShapes, SeqCombUV, SeqCombMV.

space $A$, reward function $R$, dynamics $p$, and discount $\gamma$. The goal is to learn a policy $\pi : S \rightarrow A$ that maximizes the return $\sum_{t=1}^{\infty} \gamma^{t-1} r_t$.

For a given time series input $x \in \mathbb{R}^{T \times V}$ we define the state at time $t$ as $s_t = (x, m_t)$, where $m_t \in \{0, 1\}^{T \times V}$ is a mask with exactly $t$ ones. The action at time $t$, $a_t \in \{0, 1, \ldots, T \times V - 1\}$, selects a single zero-valued index to reveal. The dynamics deterministically update the mask to $m_{t+1}$. In the next subsections, we define the reward and policy and show how they yield attribution masks.

**Reward.** As the representation space can be utilized to indicate the importance of feature masks $m$, we define the reward of each action $a_t$ as the squared distance between the current reconstructed representation and the next reconstructed representation in the latent space after revealing the chosen feature:

$$r(a_t) = \|\tilde{z}_{t+1} - \tilde{z}_t\|_2^2 \tag{3}$$

Here $\tilde{z}_t$ denotes the reconstructed embedding at state $t$, provided by mask $m_t$. Importantly, we define rewards as the *latent change* between consecutive reconstructions rather than the residual gap to the fully unmasked embedding. This local difference preserves conditional dependencies and values features by their marginal impact in context. Absolute progress toward the full embedding would bias credit toward early features and suppress synergies.

**Proposition 1 (Expected risk reduction).** Let $z = G(x)$ be the reference latent and let $I_t$ denote the information revealed by mask $m_t$. If $R_\psi$ is trained with squared loss so that $\tilde{z}_t \approx \mathbb{E}[z \,|\, I_t]$, then

$$\mathbb{E}[r(a_t) \,|\, I_t] = \mathbb{E}\big[\|z - \tilde{z}_t\|_2^2 - \|z - \tilde{z}_{t+1}\|_2^2 \,\big|\, I_t\big],$$

i.e., the *expected* latent shift equals the *expected* drop in latent MSE. Thus, summing rewards estimates the total risk reduction from the initial to the budgeted mask, yielding a principled, path-aware proxy for information gain.

*Proof sketch.* With squared loss, $\tilde{z}_t = \mathbb{E}[z|I_t]$ is the Bayes estimator. Applying the law of total variance gives the identity above by noting that the reduction in $\operatorname{tr} \operatorname{Var}(z \,|\, I)$ equals the variance of the updated posterior mean $\mathbb{E}[z|I_{t+1}]$.

*Implementation note.* We use Euclidean distances in latent space. We provide an empirical ablation in Appendix B.2.

**Policy.** We leverage the C51 algorithm (Bellemare et al., 2017) to guide the RL policy responsible for selecting which features of the time series data to unmask. C51 is a distributional RL algorithm that approximates the action-value distribution by discretizing it into 51 fixed bins and their corresponding probabilities, allowing for a detailed representation of the variability in future rewards. This is particularly advantageous in our context, where rewards can exhibit significant variability across samples. By using C51's discretization of reward distribution into bins, our model can accurately capture the uncertainty and diversity of potential outcomes associated with each feature selection prediction. This leads to robust feature attribution as the policy can better account for uncertainty in the explanatory power of the feature it selects. Consequently, C51 helps the agent make nuanced predictions about which features to prioritize, driving the reliability and interpretability of the model explanations. For more information on the C51 algorithm, see (Bellemare et al., 2017). Figure 1 illustrates the full model architecture and training pipeline.

### 3.4 OBTAINING ATTRIBUTION MAPS

For a time-series instance $x$, let $s_0 = (x, \mathbf{0})$ denote the initial state with all entries masked. RLTime produces a distributional value $Z(s_0, a)$ for each candidate reveal action $a = (t, v)$. We define saliency as the *expected return* of the initial-state actions:

$$A[t,v] = \mathbb{E}\big[Z(s_0, (t,v))\big], \quad A \in \mathbb{R}^{T \times V}.$$

We then normalize $A$ to $[0, 1]$ per instance for visualization and thresholding. Higher $A[t,v]$ indicates greater expected risk reduction if the corresponding feature $(t, v)$ is revealed.

## 4 EXPERIMENTS

**Datasets.** In this section, we evaluate the quality of our model's explanations on three synthetic datasets and four real-world datasets, following Liu et al. (2024a); Queen et al. (2023). The synthetic datasets were designed by Queen et al. (2023) and are named **FreqShapes**, **SeqComb-UV**, and **SeqComb-MV**. These datasets encapsulate a wide array of temporal dynamics within both univariate and multivariate settings. For the details of the generation of these synthetic datasets, we refer to Queen et al. (2023) Appendix C.1 and C.4. We also employ four datasets from real-world time-series classification tasks: **ECG** (Moody & Mark, 2001) - ECG arrhythmia detection; **PAM** (Reiss & Stricker, 2012) - human activity recognition; **Epilepsy** (Andrzejak et al., 2001) - EEG seizure detection; and **Boiler** (Shohet et al., 2020) - mechanical fault detection. The statistics of all the above-mentioned datasets are listed in Table 3.

Table 1: Attribution explanation performance on univariate and multivariate synthetic datasets. For all metrics, higher indicates better. Here $*$ denotes a significant difference between RLTime and best baselines over the Wilcoxon signed-rank test. First and Second place solution is highlighted in **bold** and underline font, respectively.

| Dataset | Method | Metric | | | Sum. Rank |
| | | AUPRC | AUP | AUR | |
| --- | --- | --- | --- | --- | --- |
| FreqShape | IG | 0.7516±0.0032(4) | 0.6912±0.0028(4) | 0.5975±0.0020(4) | 12 |
| | DynaMask | 0.2201±0.0013(5) | 0.2952±0.0037(5) | 0.5037±0.0015(5) | 15 |
| | TimeX | 0.8324±0.0034(3) | 0.7219±0.0031(3) | 0.6381±0.0022(3) | 9 |
| | TimeX++ | 0.8905±0.0018(2) | 0.7805±0.0014(2) | 0.6618±0.0019(2) | 6 |
| | RLTime | **1.0000**±0.0000(1)* | **0.9207**±0.0007(1)* | **0.9865**±0.0002(1)* | 3 |
| SeqCombUV | IG | 0.5760±0.0022(4) | 0.8157±0.0023(4) | 0.2868±0.0023(4) | 12 |
| | DynaMask | 0.4421±0.0016(5) | 0.8782±0.0039(3) | 0.1029±0.0007(5) | 13 |
| | TimeX | 0.7124±0.0017(3) | **0.9411**±0.0006(1) | 0.3380±0.0014(3) | 7 |
| | TimeX++ | 0.8468±0.0014(2) | 0.9069±0.0003(2) | 0.4064±0.0011(2) | 6 |
| | RLTime | **0.9549**±0.0006(1)* | 0.7609±0.0007(5) | **0.7701**±0.0012(1)* | 7 |
| SeqCombMV | IG | 0.3298±0.0015(4) | 0.7483±0.0027(4) | 0.2581±0.0028(4) | 12 |
| | DynaMask | 0.3136±0.0019(5) | 0.5481±0.0053(5) | 0.1953±0.0025(5) | 15 |
| | TimeX | 0.6878±0.0021(3) | 0.8326±0.0008(3) | 0.3872±0.0015(3) | 9 |
| | TimeX++ | 0.7589±0.0014(2) | **0.8783**±0.0007(1) | 0.3906±0.0011(2) | 5 |
| | RLTime | **0.9137**±0.0011(1)* | 0.8514±0.0010(2) | **0.5937**±0.0013(1)* | 4 |

**Baselines.** We evaluate the method against four explainability baselines, IG (Sundararajan et al., 2017), DynMask (Crabbé & Van Der Schaar, 2021), TimeX (Queen et al., 2023) and TimeX++ (Liu et al., 2024a). All reported results for our method and baselines are presented as mean ± std from a five-fold cross-validation run across three seeds. The reconstruction model $R_\psi$ is instantized as a U-Net with 1D convolutions to help maintain context and localization of the input signal. The reference model is instantiated as an encoder-decoder Transformer structure following the configuration of Liu et al. (2024a). We conduct an ablation study over the selection of the reference model in Appendix B.4 to prove that RLTime is reference model-independent.

**Metrics.** For synthetic datasets, where the precise salient features are known, we use these as the ground truth to evaluate the quality of explanations. Specifically, we consider known predictive signals within each input time series sample when interpreting a strong predictor. Following Crabbé & Van Der Schaar (2021), we assess explanation quality using the area under the precision (AUP) and area under the recall (AUR) curves. Additionally, we report the explanation area under the precision-recall curve (AUPRC), which integrates the AUP and AUR metrics. For real-world datasets,

ground truth labels for evaluating explanations are not available. In line with Queen et al. (2023), we occlude the bottom $p$-percentile of features as identified by the explainer and measure the resulting change in prediction AUROC. A robust explainer should identify key features such that when a high proportion ($p$) is occluded, the prediction performance remains stable. For all metrics, higher values indicate better explanation quality. Detailed definitions for each metric are provided in Appendix A.2. Results are averaged over 3 random seeds, each across 5 distinct data splits. The data splits are provided by previous work (Queen et al., 2023).

### 4.1 FEATURE IMPORTANCE FOR SYNTHETIC DATASETS

We conduct the feature importance evaluation on the three highlighted synthetic datasets compared against all baseline explainers in Table 1. Across all datasets, RLTime yields the best baseline explainers on 7/9 (3 metrics on 3 datasets) with an average improvement in the explanation AUPRC (15.15%), AUP (-1.42%), and AUR (63.52%) against the strongest baselines. We observe the largest improvement on AUR. This aligns with our hypothesis that relaxation-based methods are myopic (early, easy features are reinforced while synergistic ones are pruned) and that the ensuing soft-to-hard gap causes interaction breakage at evaluation. RLTime operates natively with discrete reveals, eliminating fractional leakage and preserving interactions, which improves coverage and thus AUR.

### 4.2 OCCLUSION EXPERIMENT FOR REAL-WORLD DATASETS

We conduct an occlusion experiment on real-world datasets to assess explanation faithfulness in the absence of ground-truth masks. Following prior work (Queen et al., 2023; Liu et al., 2024a), we remove the *bottom $k$-percentile* of features according to each explainer and measure downstream performance (AUROC/AUPRC). A faithful explainer should identify low-salience regions such that removing them leaves performance stable (or slightly improved by denoising). Figure 4 shows that RLTime matches or exceeds baselines across both univariate (Epilepsy, MITECG) and multivariate (PAM, Boiler) settings, with particularly strong stability (narrower error bars). We also observe several interesting scenarios. Specifically, RLTime is marginally outperformed by TimeX++ on Boiler and MITECG when the number of features remain low. However, RLTime remains the best-performed one under extreme cases, e.g. remain 99% features.

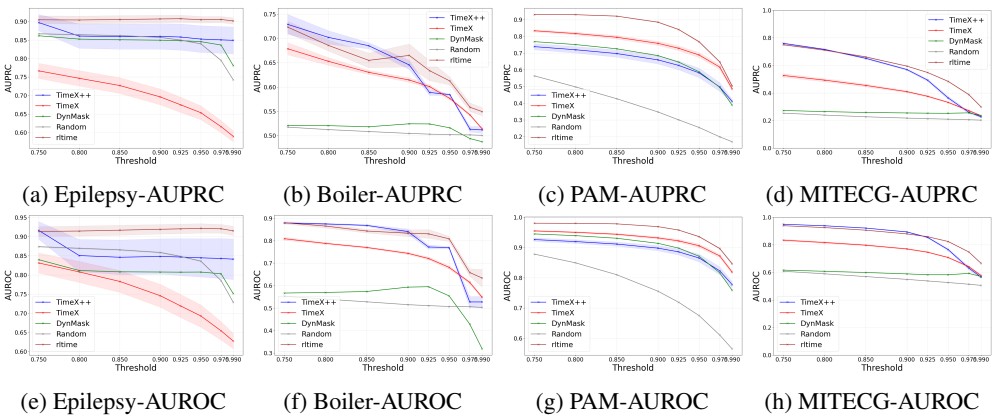

Figure 4: Occlusion Experiment on Real-world Datasets. In all figures, higher indicates better.

Complementarily, we evaluate a *deletion* test: we remove the *top* 10% of salient features and substitute them either with the per-feature mean or zeros Liu et al. (2024a). We report the post-mask predictive performance—lower numbers indicate larger degradation and hence stronger explanations. Results on all four datasets (Table 7) show that RLTime consistently achieves the strongest deletion effect, with the zero substitution typically yielding the largest performance drop. The only exception happens on the Boiler dataset, where TimeX achieves the lowest AUROC.

## 4.3 INVESTIGATION OVER THE SEARCH SPACE

To help understand the performance boosts provided by our proposed RLTime, we conduct further investigation into the search space. Here, instead of utilizing the reconstruction model as the backbone for the RL environment, we directly utilize the reconstruction model to obtain the feature attributions of the reference model. To do this, we define the importance of a feature as the

$$A_{t,v} = \|\tilde{z}_{t,v} - E(R_\psi(x, \mathbf{0}))\|_2^2, \tag{4}$$

where $A_{t,v}$ indicates the feature attribute at time step $t$ and variable $v$. $\mathbf{0}$ indicates a fully masked tensor with all values equal to zero. $\tilde{z}_{t,v}$ indicates the latent representation given mask $m_{t,v}$, which indicates a fully masked tensor with only $(t, v)$ set to 1. The pseudo-algorithm is shown in Algorithm 1. This algorithm provides a strong *one-step set score* at the initial state—no feature unmasked. It is *not identical* to our RL reward (which uses the latent *shift* between consecutive reconstructions), but serves as a closely related, myopic baseline. We refer to it as *Naive Search*.

The detailed comparison between Naive Search and RLTime is conducted over FreqShape, SeqCombUV, and SeqCombMV datasets. The results are listed in Table 2. Apart from listing the performance of both methods, we further listed the inference speeds. The following observations can be made.

First, Naive Search, although being a simple and straightforward solution for feature attribution, proves to be a very strong baseline. It outperforms all existing baselines, such as TimeX and TimeX++, with the performances listed in Table 1. This is consistent with the **soft-to-hard gap** as relaxation-based explainers train with fractional masks but are evaluated with sparse, discrete selections, which breaks interactions. In contrast, Naive Search scores discrete one-step reveals using the latent-reconstruction metric. Because the reconstruction model was trained on many partial masks, it already encodes effects of features and their dependencies; enumerating actions at the initial state can therefore yield high-quality attributions without incurring the soft-to-hard mismatch.

Second, Naive Search, while removing the soft-to-hard gap underperforms the proposed RLTime in terms of overall performance. We hypothesize its limitation is **myopic exploration**: scoring actions only from the initial mask cannot credit features whose contribution is conditional on information revealed later, which limits coverage. RLTime learns a distributional value function over discrete reveals that evaluates actions under the current revealed set, preserving interactions and exploring the space more fully. This can be observed with the performance gap on AUPRC, AUP, and AUR between these two solutions. We can easily observe that RLTime can improve up to ∼60% in terms of AUR compared to Naive Search, reflecting its increasing ability in exploration. In comparison, RLTime only boosts the performance of AUPRC by up to ∼5% in terms of AUPRC.

Finally, Naive Search is significantly slower at inference, since it requires a reconstruction forward pass for every candidate action to compute scores. Once trained, RLTime selects features with a single value-function evaluation per step (with action masking), yielding much faster inference.

Table 2: Performance comparison between Naive Search and RLTime. For all metrics except Inference(s), higher indicates better.

| Dataset | Metric | Method | | Improvement | |
|---|---|---|---|---|---|
| | | Naive Search | RLTime | Absolute | Relative(%) |
| FreqShape | AUPRC↑ | 0.9993±0.0002 | 1.0000±0.0000 | +0.0007 | 0.07% |
| | AUP↑ | 0.9381±0.0004 | 0.9207±0.0007 | −0.0174 | −1.85% |
| | AUR↑ | 0.9366±0.0006 | 0.9865±0.0002 | +0.0499 | 5.33% |
| | Inference(s)↓ | 1.64±0.12 | 0.06±0.07 | 1.58 | 96.34% |
| SeqCombUV | AUPRC↑ | 0.9028±0.0010 | 0.9549±0.0006 | +0.0521 | 5.77% |
| | AUP↑ | 0.9722±0.0002 | 0.7609±0.0007 | −0.2113 | −21.74% |
| | AUR↑ | 0.4408±0.0017 | 0.7701±0.0012 | +0.3293 | 74.70% |
| | Inference(s)↓ | 19.93±0.06 | 0.05±0.07 | 19.88 | 99.75% |
| SeqCombMV | AUPRC↑ | 0.8897±0.0007 | 0.9137±0.0011 | +0.0240 | 2.70% |
| | AUP↑ | 0.9795±0.0002 | 0.8514±0.0010 | −0.1281 | −13.08% |
| | AUR↑ | 0.4648±0.0014 | 0.5937±0.0013 | +0.1289 | 27.73% |
| | Inference(s)↓ | 60.49±0.02 | 0.06±0.06 | 60.43 | 99.90% |

## 5 RELATED WORK

**Time-series Explainability.** Recent advancements in deep learning for time series have introduced various methods to improve explainability Rojat et al. (2021), primarily focusing on identifying key segments of time-series data that significantly impact model predictions. These approaches often operate in a post-hoc manner, attempting to explain already trained models. Attention-based approaches, such as Lin et al. (2020); Choi et al. (2016), utilize attention mechanisms to generate scores that are inherently linked to the model coefficients. Perturbation methods, such as Crabbé & Van Der Schaar (2021); Enguehard (2023); Liu et al. (2024b); Bento et al. (2021), offer interpretability by modifying non-salient features to evaluate their effect on the model's output. These methods analyze the stability of predictions when certain features are masked, revealing the significance of specific time-series segments. Gradient-based approaches like Sundararajan et al. (2017) go beyond perturbations, using gradients to attribute input feature importance. IG adheres to key principles such as *Sensitivity* and *Implementation Invariance*, ensuring that important features are identified consistently across models. Surrogate-based solutions tend to map the prediction results into some white-box models, such as linear model (Ribeiro et al., 2016). Our approaches also belongs to surrogate-based solutions. It builds upon existing works (Queen et al., 2023; Liu et al., 2024a; Yue et al., 2025) which aim to provide more inherent interpretability within time-series models via training additional complex component specializing in providing explainability. For instance, Queen et al. (2023) trains an interpretable surrogate to mimic the reference model by introducing model behavior consistency, aiming to preserves relations in the latent space. Liu et al. (2024a) addresses a potential *signaling issue* that arises in explainability models that rely on information bottlenecks. Yue et al. (2025) adopts similar architecture but instead utilize conditional mutual information to minimize redundancy and maximize completeness as optimization objectives. In these models, the reconstruction or label-matching approaches can result in artificial signals, where outputs are manipulated to match predictions without accurately reflecting the underlying processes. This concern is particularly relevant for models attempting to mimic the behavior of reference models via indirect signals, rather than genuine understanding.

**Reinforcement Learning (RL) for Explainability.** Reinforcement learning has been widely utilized in providing explainability in various domains. INVASE (Yoon et al., 2018) introduces a selector network trained using an actor-critic framework, aiming to minimize the loss difference between a predictor network and a baseline network while enforcing sparsity through an $\ell_0$ penalty. ASAC (Yoon et al., 2019) extends this idea to a medical context, using actor-critic methods to determine which observations should be obtained next, optimizing the information gained from limited interactions with the environment. Similarly, Active MR (Pineda et al., 2020) addresses the problem of MRI acquisition by modelling it as a Partially Observable Markov Decision Process (POMDP). Their approach leverages deep RL to optimize the sequence of measurements, balancing reconstruction quality and acceleration factors for the MRI machine. In the context of real-time explainability, CoRTX (Chuang et al., 2023) proposes a Real-time Explainer (RTX) model. This method generates explanations in a single forward pass of an explanation network, ensuring efficiency in high-stakes scenarios where explanations must be produced quickly. Finally, Learning to Explain (L2X) (Chen et al., 2018) introduces an instance-level explainer model that optimizes a lower bound on mutual information. L2X generates explanations by learning to select a subset of input features that carry the most information about the prediction, offering a flexible and data-driven approach to explainability. RLTime follows previou works by extending the utilization of reinforement learning into time series explainability domain.

## 6 CONCLUSION

In this paper, we propose the RLTime framework, a novel approach to time series explainability by reframing the problem as learning through sequential information acquisition. This RLTime method overcomes key limitations of prior techniques, such as the myopic exploration of feature space and the soft-to-hard gap that leads to misranking of features at evaluation. Extensive empirical evidence against SOTA baselines on both synthetic and real-world datasets proves the effectiveness of RLTime, especially in exploration-exploitation balancing. Further investigation on the environment using naive search demonstrates the effectiveness of the masked reconstruction models.

## REPRODUCIBILITY STATEMENT

We place a strong emphasis on reproducibility. To ensure reproducibility of our paper, we disclose all the utilized hyperparameters in Appendix A.4 with implementation details carefully described in Appendix A.5. The hyperparameters for the reference model are also provided in Appendix A.3. The implementation of our code is going through the internal approval process. We promise to open-source the source code and all related details after the internal approval.

## ETHIC STATEMENT

This paper does not involve any artifact that directly involves ethical consideration. Generally, explaining the behavior of models helps users to better understand these behaviors and grants them more control.

## LLM USAGE

LLM is generally used as a polishing tool during the paper-writing process. LLM usage is not involved in idea formulation and draft writing.

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

## A   ADDITIONAL EXPERIMENTAL SETUP DETAILS

### A.1   STATISTICS OF DATASETS

Here, we further list the statistics of all the datasets in Table 3.

Table 3: Dataset Statistics

| Dataset | #Samples($N$) | Length($T$) | #Variables($V$) | Classes($C$) |
|---|---|---|---|---|
| FreqShapes | 6,100 | 50 | 1 | 4 |
| SeqCombUV | 6,100 | 200 | 1 | 4 |
| SeqCombMV | 6,100 | 200 | 4 | 4 |
| MITECG | 92,511 | 360 | 1 | 5 |
| PAM | 5,333 | 600 | 17 | 8 |
| Epilepsy | 11,500 | 178 | 1 | 2 |
| Boiler | 160,719 | 36 | 20 | 2 |

### A.2   DEFINING AUPRC, AUR, AUP

Following Crabbé & Van Der Schaar (2021), we adopt **AUP** and **AUR** as evaluation metrics for the quality of feature attribution maps. Let $Q \in \{0,1\}^{T \times V}$ be the ground-truth where each element indicates whether the corresponding feature is salient in $x \in \mathbb{R}^{T \times V}$. Let $A \in [0,1]^{T \times V}$ be an attribution map obtained from certain explanation methods. Then given a detection threshold $\tau \in (0,1)$, we are able to obtain an estimation $\hat{Q}_{t,v}(\tau)$ via:

$$\hat{Q}_{t,v}(\tau) = \begin{cases} 1 & \text{if } A_{t,v} > \tau, \\ 0 & \text{otherwise.} \end{cases}$$

Considering the set of true saliency indexes and indexes selected by the estimator:

$$Q = \{(t,v) \in [1:T] \times [1:V] \mid Q_{t,v} = 1\}$$

$$\hat{Q}(\tau) = \left\{(t,v) \in [1:T] \times [1:V] \mid \hat{Q}_{t,v}(\tau) = 1\right\},$$

as well as the precision and recall curves, we can map each threshold to a precision and recall score:

$$\mathbf{P} : (0,1) \longrightarrow [0,1] : \tau \longmapsto \frac{|Q \cap \hat{Q}(\tau)|}{|\hat{Q}(\tau)|},$$

$$\mathbf{R} : (0,1) \longrightarrow [0,1] : \tau \longmapsto \frac{|Q \cap \hat{Q}(\tau)|}{|Q|}.$$

Therefore, the AUP and AUR scores can be calculated as:

$$\text{AUP} = \int_0^1 \text{P}(\tau)d\tau$$

$$\text{AUR} = \int_0^1 \text{R}(\tau)d\tau$$

As for the AUPRC value, we compute it jointly with AUP and AUR as:

$$\text{AUPRC} = \int_0^1 \text{P}(\text{R})d\text{R}$$

### A.3 HYPERPARAMETERS FOR REFERENCE MODEL

For the vanilla Transformer, we report the detailed hyperparameters for its training in Table 4, which overlaps with Queen et al. (2023) and Liu et al. (2024a) to ensure the fairness of the comparison in explanation. The classification performance of each transformer over all datasets is also shown in Table 5. We observe that the reference models perform well on all datasets, ensuring a good foundation for further explainability studies (Faber et al., 2021).

Table 4: Training parameters for reference models across all datasets

| Parameter | Freqshape | SeqCombUV | SeqCombMV | Epilepsy | Boiler | PAM | MITECG |
|---|---|---|---|---|---|---|---|
| lr | 1e-3 | 1e-3 | 5e-4 | 1e-4 | 2e-3 | 1e-3 | 5e-4 |
| l2 | 1e-1 | 1e-2 | 1e-3 | 1e-3 | 1e-3 | 1e-2 | 1e-3 |
| epochs | 100 | 200 | 1000 | 300 | 1000 | 100 | 1000 |
| # layers | 1 | 2 | 2 | 1 | 1 | 1 | 1 |
| # hear | 1 | 1 | 1 | 1 | 1 | 1 | 1 |
| $d_h$ | 16 | 64 | 128 | 16 | 32 | 72 | 32 |
| dropout | 0.1 | 0.25 | 0.25 | 0.1 | 0.25 | 0.25 | 0.1 |
| norm embedding | False | False | False | False | True | False | True |

Table 5: Classification performance of transformer-based reference models across all datasets.

| Dataset | F1 | AUPRC | AUROC |
|---|---|---|---|
| FreqShapes | 0.9891±0.0031 | 0.9999±0.0001 | 0.9998±0.0001 |
| SeqCombUV | 0.9356±0.0101 | 0.9798±0.0078 | 0.9912±0.0032 |
| SeqCombMV | 0.9742±0.0031 | 0.9953±0.0008 | 0.9987±0.0012 |
| Epilepsy | 0.9201±0.0023 | 0.9218±0.0041 | 0.9354±0.0056 |
| Boiler | 0.8421±0.0091 | 0.8145±0.0102 | 0.8987±0.0087 |
| PAM | 0.8945±0.0078 | 0.9301±0.0093 | 0.9765±0.0079 |
| MITECG | 0.9012±0.0256 | 0.9311±0.0251 | 0.9423±0.0228 |

### A.4 HYPERPARAMETERS FOR RLTIME

In this section, we list all the detailed hyperparameters during the training process of RLTime in Table 6. Here *unmask range* indicates the maximum unmask length for the reconstruction model, where, during the training time, masked features are randomly selected.

Table 6: Training parameters for the proposed RLTime across all datasets. The upper part refers to the hyperparameters during the training of the reconstruction environment. The bottom part refers to the hyperparameters during the training of the C51 model.

| Parameter | Freqshape | SeqCombUV | SeqCombMV | Epilepsy | Boiler | PAM | MITECG |
|---|---|---|---|---|---|---|---|
| lr | 3e-4 | 3e-4 | 3e-4 | 3e-2 | 3e-4 | 1e-1 | 3e-4 |
| l2 | 1e-4 | 0 | 1e-4 | 1e-6 | 1e-5 | 1e-2 | 1e-5 |
| unmask range | 20 | 30 | 100 | 30 | 50 | 10000 | 20 |
| epochs | 300 | 300 | 300 | 300 | 300 | 300 | 300 |
| $\lambda_X$ | 1.0 | 1.0 | 1.0 | 1.0 | 1.0 | 1.0 | 1.0 |
| $\lambda_Z$ | 0.1 | 0.1 | 1.0 | 0.1 | 1.0 | 0.1 | 0.1 |
| $\lambda_P$ | 0.1 | 0.1 | 0.5 | 0.1 | 1.0 | 0.1 | 1.0 |
| $\epsilon$ | 1e-3 | 1e-4 | 1e-4 | 1e-4 | 1e-4 | 1e-5 | 1e-3 |
| $\gamma$ | 0.1 | 0.3 | 0.1 | 0.1 | 0.1 | 0.9 | 0.1 |
| buffer size | 1e+6 | 1e+6 | 3e+6 | 3e+6 | 3e+5 | 1e+5 | 5e+5 |
| time steps | 1e+6 | 1e+6 | 1e+7 | 1e+7 | 1e+6 | 1e+6 | 1e+6 |
| horizon | 4 | 10 | 2 | 10 | 40 | 6 | 2 |
| # atoms | 121 | 201 | 201 | 201 | 51 | 11 | 201 |
| v_min | -0.2 | -1.0 | -1.0 | -1.0 | -1.0 | -1.0 | -1.0 |
| v_max | +1.0 | +1.0 | +1.0 | +1.0 | +1.0 | +1.0 | +1.0 |

## A.5 IMPLEMENTATION DETAILS

We compare our RLTime model and environment against several implementations. For TimeX (Queen et al., 2023) and TimeX++ (Liu et al., 2024a), we adopt the official implementations, respectively[12]. For IG (Sundararajan et al., 2017) and Dynamask (Crabbé & Van Der Schaar, 2021), we reuse the 3rd-party implementation provided by TimeX++. All hyperparameters of the specific models follow the code provided by the corresponding papers.

## A.6 PSEUDO CODE FOR OBTAINING FEATURE ATTRIBUTION VIA NAIVE SEARCH

In this section, we append the pseudo code utilized in Section 4.3 to provide a better understanding. This pseudo code is equivalent to naively enumerating all actions and computing their reward at the initial stage – no feature unmasked. Hence, we refer to this algorithm as *Naive Search*.

---
**Algorithm 1** Pseudo Algorithm for Feature Attribution
---
**Require:** data instance $x$, Conditional Generation Model $R$
**Ensure:** Feature Attribute $Z$
  1: Initialize Feature Attribute $Z$ = zeros_like($x$)
  2: **for** $p$ in #Time Steps **do**
  3:     **for** $v$ in #Variable **do**
  4:        initialize a zero mask $m$
  5:        unmask the $v$ variable at time step $t$ by setting $m_{t,v} = 1$
  6:        obtain the feature attribute $A_{t,v}$ given Eq. 4
  7:     **end for**
  8: **end for**
  9: Normalize $A$ over both dimensions $T$ and $V$ between $[0, 1]$ to obtain attribution maps
---

Table 7: Deletion test on four real-world datasets by masking the top 10% of salient features. The masked portion is substituted with the per-feature mean or with zeros. *Numbers are post-mask predictive performance* (AUROC); lower indicates a larger degradation and thus a stronger explainer. First and second place solution is highlighted in **bold** and underline font, respectively.

| Method | Substitution | MITECG | PAM | Epilepsy | Boiler | Rank |
|---|---|---|---|---|---|---|
| Dynamask | mean | 0.8287±0.0094 | 0.7221±0.0092 | 0.8553±0.0037 | 0.6558±0.0364 | 28 |
| | zero | 0.8363±0.0100 | 0.7240±0.0089 | 0.8564±0.0037 | 0.8244±0.0177 | 32 |
| TimeX | mean | 0.7351±0.0084 | 0.7118±0.0117 | 0.8115±0.0243 | 0.4908±0.0060 | 15 |
| | zero | 0.7352±0.0088 | 0.7097±0.0124 | 0.8169±0.0240 | **0.4778±0.0164** | 13 |
| TimeX++ | mean | 0.7275±0.0032 | 0.7112±0.0092 | 0.8150±0.0016 | 0.5037±0.0041 | 18 |
| | zero | 0.7051±0.0055 | 0.7165±0.0106 | 0.8424±0.0088 | 0.4968±0.0076 | 17 |
| RLTime | mean | 0.6831±0.0109 | 0.7055±0.0141 | 0.8290±0.0038 | 0.4976±0.0032 | 13 |
| | zero | **0.6732±0.0092** | **0.6934±0.0123** | **0.8102±0.0089** | 0.5011±0.0013 | 8 |

# B    ADDITIONAL EXPERIMENTS AND ABLATION STUDIES

## B.1    OCCLUSION EXPERIMENTS (CONT.)

## B.2    ABLATION ON DIFFERENT REWARD FOR RL TRAINING

In this section, we further investigate the selection of reward design during the RL training. We compare our current reward design with two different setups, shown in Equation B.2. Here *format 1* is similar to our default reward except that it is calculated in the input space, instead of the embedding space of the reference model. *format 2* and *3* adopt a different idea by computing the reward based on the distance between the next stage and the initial stage, denoted as $\mathbf{0}$. They differ from each other only in whether adopting the normalization using the latent distance between fully unmasked point $E(x)$ and fully masked point $E(R_\psi(x, \mathbf{0}))$.

$$\text{format 1} = \|R_\psi(x, m_{t+1}) - R_\psi(x, m_t))\|_2^2$$
$$\text{format 2} = \|E(R_\psi(x, m_{t+1})) - E(R_\psi(x, \mathbf{0}))\|_2^2$$
$$\text{format 3} = \frac{\|E(R_\psi(x, m_{t+1})) - E(R_\psi(x, \mathbf{0}))\|_2^2}{\|E(x) - E(R_\psi(x, \mathbf{0}))\|_2}$$

The ablation study is conducted over SeqCombUV, SeqCombMV, and Epilepsy datasets, with the result being shown in Table 8 and Figure 5. Based on the results, we can observe that the current default reward designs consistently yield SOTA across all three datasets.

Table 8: Ablation over Reward in RL Algorithm. Best-performed solution is highlighted in **bold** font.

| Dataset | Method | AUPRC | AUP | AUR |
|---|---|---|---|---|
| SeqCombUV | default | **0.9549±0.0006** | **0.7609±0.0007** | **0.7701±0.0012** |
| | format 1 | 0.9199±0.0022 | 0.7150±0.0015 | 0.7189±0.0024 |
| | format 2 | 0.8062±0.0028 | 0.6753±0.0022 | 0.5855±0.0027 |
| | format 3 | 0.8002±0.0033 | 0.6989±0.0023 | 0.5455±0.0028 |
| SeqCombMV | default | **0.9137±0.0011** | **0.8514±0.0010** | 0.5937±0.0013 |
| | format 1 | 0.8515±0.0040 | 0.6959±0.0018 | **0.6557±0.0037** |
| | format 2 | 0.7655±0.0048 | 0.6614±0.0039 | 0.5810±0.0045 |
| | format 3 | 0.7633±0.0045 | 0.7760±0.0038 | 0.4408±0.0035 |

## B.3    ABLATION ON DIFFERENT RL POLICY

In this section, we aim to justify the selection of the C51 algorithm. We fix all other components except for the RL algorithm itself and replace C51 with DQN. The experiments are conducted over

---

[1]https://github.com/mims-harvard/TimeX

[2]https://github.com/zichuan-liu/TimeXplusplus

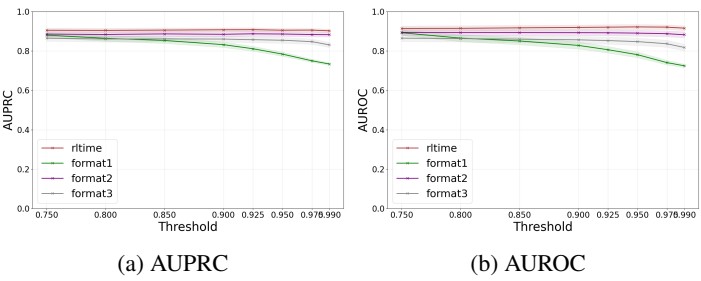

(a) AUPRC

(b) AUROC

Figure 5: Ablation Experiment on Reward over Epilepsy

the SeqCombUV, SeqCombMV, and Epilepsy datasets. C51 differs from the DQN in that it learns the full probability distribution of the cumulative return, instead of learning the expectation of the cumulative return directly. The ablation results are shown in Table 9. We can make the following observations. First, C51 generally outperforms DQN over all metrics. This may be attributed to the richer gradients introduced by the C51. Second, the divergence of C51 over instances tends to be smaller than that of the DQN, aligning with the original observations that C51 tends to improve stability than DQN in a stochastic environment (Bellemare et al., 2017). Further occlusion experiment shown in Figure 6 confirms that the DQN underperforms C51 in identifying the important features in the occlusion experiment. However, DQN shows similar performance to TimeX++ and remains a strong baseline. This further highlights the importance of utilizing a reconstruction model for creating the environments.

Table 9: Ablation over RL Algorithm. Best-performed solution is highlighted in **bold** font.

| Dataset | Method | AUPRC | AUP | AUR |
|---|---|---|---|---|
| SeqCombUV | DQN | 0.9346±0.0020 | 0.7157±0.0016 | 0.6560±0.0024 |
| | C51 | **0.9549**±0.0006 | **0.7609**±0.0007 | **0.7701**±0.0012 |
| SeqCombMV | DQN | 0.8700±0.0025 | 0.6381±0.0025 | **0.6321**±0.0023 |
| | C51 | **0.9137**±0.0011 | **0.8514**±0.0010 | 0.5937±0.0013 |

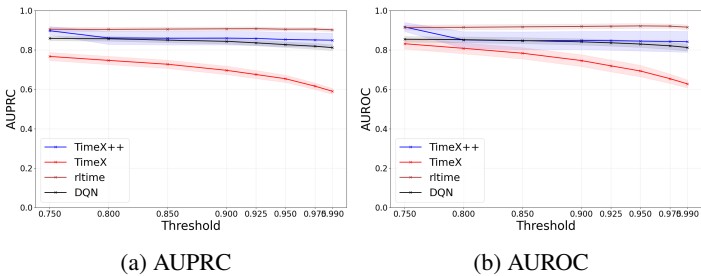

(a) AUPRC

(b) AUROC

Figure 6: Ablation Experiment on RL Algorithm over Epilepsy

### B.4 ABLATION ON DIFFERENT REFERENCE MODEL

In all previous experiments, we utilized Transformer-based time series classifiers as reference models. To showcase the flexibility of RLTime, we switch the Transformer with two classifiers following previous works Queen et al. (2023); Liu et al. (2024a) on both SeqCombUV and SeqCombMV datasets: (I) a long-short term memory (LSTM)-based classifier using a 3-layer bidirectional LSTM plus an MLP over the last hidden states, and (ii) a convolutional neural network(CNN)-based classifier using 3-layer CNNs and one MLP with mean-pooling. Table 10 and 11 respectively summarize the results over all baselines.

We consistently observe that RLTime yields the best-performing solutions compared to all existing explainers, such as IG or TimeX++. This observation is consistent with that in Table 1. Similarly, RLTime can better balance exploration-exploitation compared with Naive Search. Specifically,

RLTime significantly outperforms Naive Search in terms of AUR while underperforming in terms of AUP. RLTime tends to slightly outperform Naive Search in terms of AUPRC, which is similar to the observation in Table 4. Hence, we can conclude that RLTime and its simplified variant, Naive Search, are agnostic to the reference models.

Table 10: Explainer results with CNN predictor on SeqComb-UV and SeqComb-MV datasets. First and second place solution is highlighted in **bold** and underline font, respectively.

| Dataset | Method | Metric | | | Sum Rank |
|---|---|---|---|---|---|
| | | AUPRC | AUP | AUR | |
| SeqCombUV | IG | 0.5865±0.0014 | 0.8234±0.0012 | 0.2901±0.0029 | 15 |
| | DynaMask | 0.4512±0.0013 | 0.8790±0.0023 | 0.1046±0.0009 | 14 |
| | TimeX | 0.7298±0.0016 | 0.8237±0.0012 | 0.4112±0.0011 | 11 |
| | TimeX++ | 0.8332±0.0015 | 0.8765±0.0016 | 0.3921±0.0009 | 10 |
| | Naive Search | 0.9356±0.0014 | **0.9845**±0.0018 | 0.4700±0.0023 | 5 |
| | RLTime | **0.9538**±0.0015 | 0.7544±0.0016 | **0.7751**±0.0020 | 8 |
| SeqCombMV | IG | 0.5979±0.0027 | 0.8858±0.0014 | 0.2294±0.0013 | 14 |
| | DynaMask | 0.4550±0.0016 | 0.7308±0.0025 | 0.3135±0.0019 | 17 |
| | TimeX | 0.7016±0.0019 | 0.7670±0.0012 | 0.4689±0.0016 | 11 |
| | TimeX++ | 0.7822±0.0012 | 0.8896±0.0005 | 0.3434±0.0012 | 9 |
| | Naive Search | 0.8941±0.0015 | **0.9789**±0.0006 | 0.7035±0.0023 | 5 |
| | RLTime | **0.8990**±0.0043 | 0.7486±0.0024 | **0.7593**±0.0034 | 7 |

Table 11: Explainer results with LSTM predictor on SeqComb-UV and SeqComb-MV datasets. First and second place solution is highlighted in **bold** and underline font, respectively.

| Dataset | Method | Metric | | | Sum Rank |
|---|---|---|---|---|---|
| | | AUPRC | AUP | AUR | |
| SeqCombUV | IG | 0.5721±0.0016 | 0.8154±0.0020 | 0.2986±0.0019 | 15 |
| | DynaMask | 0.4437±0.0011 | 0.8776±0.0013 | 0.1066±0.0014 | 16 |
| | TimeX | 0.7048±0.0013 | 0.9037±0.0014 | 0.3203±0.0012 | 10 |
| | TimeX++ | 0.8412±0.0012 | 0.8851±0.0011 | 0.4121±0.0009 | 9 |
| | Naive Search | 0.9361±0.0014 | **0.9845**±0.0002 | 0.4549±0.0024 | 5 |
| | RLTime | **0.9493**±0.0015 | 0.7424±0.0017 | **0.7741**±0.0018 | 8 |
| SeqCombMV | IG | 0.2369±0.0020 | 0.5150±0.0048 | 0.3211±0.0032 | 15 |
| | DynaMask | 0.2836±0.0021 | 0.6369±0.0047 | 0.1816±0.0015 | 14 |
| | TimeX | 0.1298±0.0017 | 0.1307±0.0022 | 0.4751±0.0015 | 14 |
| | TimeX++ | 0.4052±0.0038 | 0.6804±0.0052 | 0.3519±0.0021 | 10 |
| | Naive Search | 0.9034±0.0015 | **0.9939**±0.0036 | 0.3904±0.0019 | 6 |
| | RLTime | **0.9112**±0.0021 | 0.8192±0.0017 | **0.5901**±0.0029 | 4 |

## B.5 FURTHER COMPARISON WITH SOTA BASELINE ORTE

Apart from TimeX and TimeX++, one recent work, namely ORTE (Yue et al., 2025), has also been proposed and evaluated under similar setups. However, by the time of submission, the source code of ORTE had not been released[3]. Hence, we directly copy their results with numerical values here to conduct a rough comparison. With the results shown in Table 12, we can have similar observations in Section 4.1 when comparing ORTE with RLTime.

## B.6 ABLATION ON TRAINING TIME EFFICIENCY

In this section, we further evaluate the train time efficiency of our proposed RLTime versus other methods requiring model training, namely TimeX (Queen et al., 2023) and TimeX++ (Liu et al., 2024a) Specifically, we report the training time on SeqCombUV, SeqCombMV and Epilepsy datasets in Table 13.

---

[3]https://github.com/moon2yue/ORTE_public

Table 12: Attribution explanation performance between ORTE and RLTime.

| Dataset | Method | Metric | | |
|---|---|---|---|---|
| | | AUPRC | AUP | AUR |
| FreqShape | ORTE | 0.9998±0.0001 | 0.8269±0.0014 | 0.8298±0.0020 |
| | RLTime | **1.0000**±0.0000 | **0.9207**±0.0007 | **0.9865**±0.0002 |
| SeqCombUV | ORTE | 0.9001±0.0025 | **0.9711**±0.0006 | 0.4503±0.0031 |
| | RLTime | **0.9549**±0.0006 | 0.7609±0.0007 | **0.7701**±0.0012 |
| SeqCombMV | ORTE | 0.8314±0.0019 | **0.9011**±0.0005 | 0.5632±0.0028 |
| | RLTime | **0.9137**±0.0011 | 0.8514±0.0010 | **0.5937**±0.0013 |

Table 13: Training Efficiency compared against TimeX and TimeX++. All values are measured in second.

| | Method | SeqCombUV | SeqCombMV | Epilepsy |
|---|---|---|---|---|
| | TimeX | 232.9 | 469.5 | 619.5 |
| | TimeX++ | 312.3 | 454.2 | 594.3 |
| RLTime | Reconstruction | 321.2 | 412.2 | 522.1 |
| | Reinforcement | 284.3 | 313.1 | 784.7 |
| | Total | 605.5 | 725.3 | 1306.8 |

## B.7 HYPERPARAMETER SENSITIVITY STUDY

In this section, we aim into investigate the role of different hyperparameters during the training of RLTime, incluing both masked reconstruction model and reinforcement learning model. Specifically, we evaluate both the learning rate, L2-regularization and unmasked range for masked reconstruction model, as well as $\gamma$ and max horizon for reinforcement learning model. Based on the result shown in Figure 7, we can observe that both models are robust against hyperparameters. In comparison, the reinforcement learning model is more sensitive compared to the masked reconstruction model.

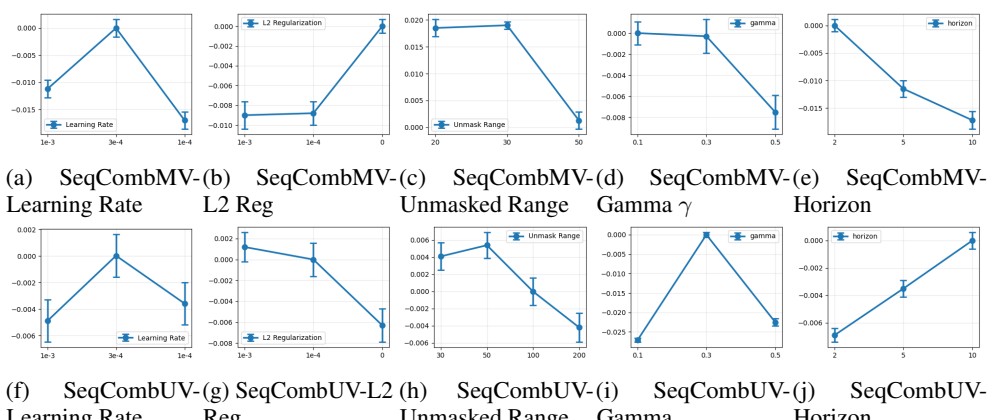

(a) SeqCombMV-Learning Rate (b) SeqCombMV-L2 Reg (c) SeqCombMV-Unmasked Range (d) SeqCombMV-Gamma $\gamma$ (e) SeqCombMV-Horizon

(f) SeqCombUV-Learning Rate (g) SeqCombUV-L2 Reg (h) SeqCombUV-Unmasked Range (i) SeqCombUV-Gamma (j) SeqCombUV-Horizon

Figure 7: Hyperparameter Study on SeqCombUV and SeqCombMV datasets. Performance gap against the default setting is reported.

## C VISUALIZATION

Saliency maps are considered to be an important tool for visualizing the significance of different features, particularly in multivariable time series analysis (Queen et al., 2023; Liu et al., 2024a). Hence, we demonstrate the saliency maps of different baselines and our proposed solutions, both Naive Search (shortened as Naive) and RLTime, on FreqShape, SeqCombUV, and SeqCombMV datasets. The visualization results over all three datasets are shown in Figure 8, 9, and 10, respectively. For all figures, we visualize five random data instances over all explainers, with the ground truth

explanations provided in the bottom row. We can make the following observations based on the visualization results:

- We can easily observe that across all three datasets, IG tends to identify a large number of features as important, with wider and more diffuse shaded areas. It seems to identify the general vicinity of the important features, but is less precise than other methods in pinpointing the exact critical time steps.

- We can also observe that on the SeqCombMV and FreqShape datasets, Dynamask tends to ignore important key features, often detecting only one or two salient segments with erroneous values. This correlates with the low-AUP value of Dynamask in Table 1. In contrast, on the SeqCombUV dataset, Dynamask is able to cover many important features in its saliency results. However, it seems to identify the general vicinity of the important features, yielding more diffuse shaded areas. This also correlates with the low AUR value in Table 1.

- TimeX and TimeX++ appear to be more precise, as they produce explanations that are visually narrower and more focused, often aligning closely with the ground truth shaded areas. TimeX++, in particular, seems to have a strong performance, as its shaded regions consistently match the location and width of the ground truth more accurately than the others. This suggests that TimeX++ is a highly effective model for this specific task and dataset. However, both methods still surfer from the two drawbacks, namly myopic-exploration and soft-to-hard gap, identified in the introduction section. It appears that both TimeX and TimeX++ may either myopicially missing several important positions or assign small importance values on many positions, leading to the soft-to-hard gap during discretization.

- In stark contrast, both of our proposed methods, namely RLTime and Naive (Search), focus on the peak points in the time series for matching to ground-truth explanations. However, Naive Search and RLTime still have their own difference, as RLTime tends to be more explorative and Naive Search tends to be more exploitative. This can be reflected in the generated saliency, which tends to have wider scopes.

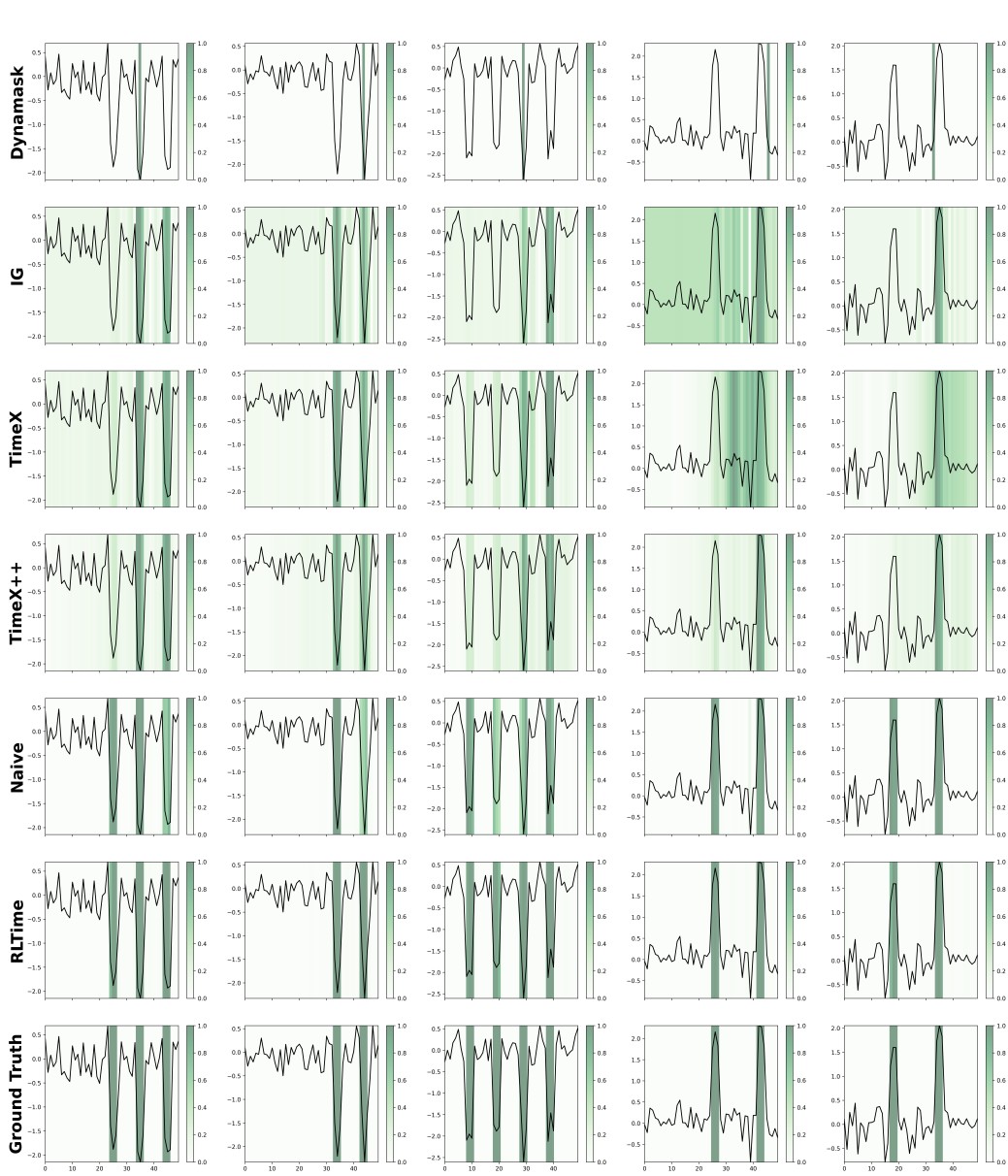

Figure 8: Visualization of all explainers on the FreqShapes dataset. Each column corresponds to a unique and randomly selected sample. For each row, the method used to generate the corresponding explanation figure is indicated, with the ground truth explanations presented in the bottom row.

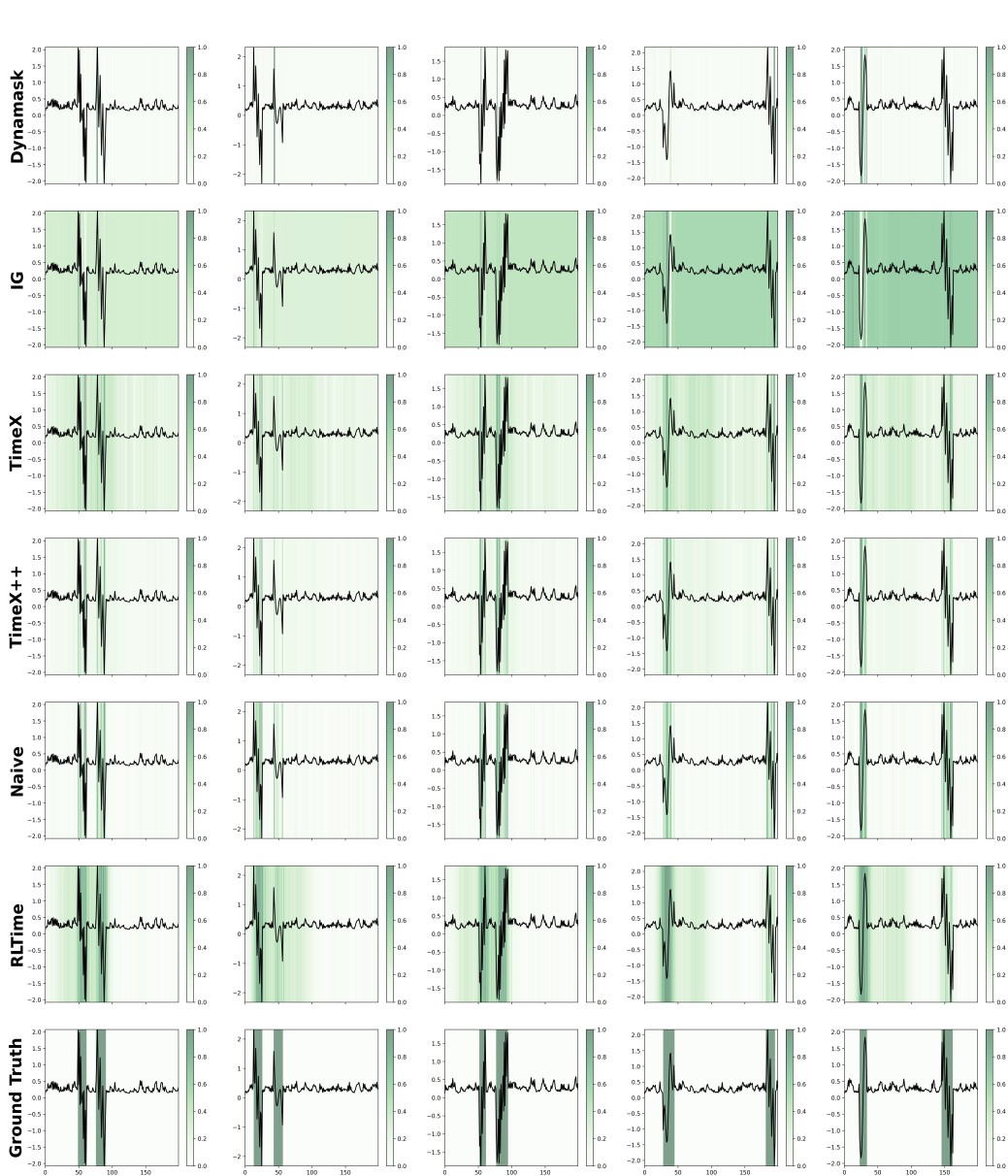

Figure 9: Visualization of all explainers on the SeqCombUV dataset. Each column corresponds to a unique and randomly selected sample. For each row, the method used to generate the corresponding explanation figure is indicated, with the ground truth explanations presented in the bottom row.

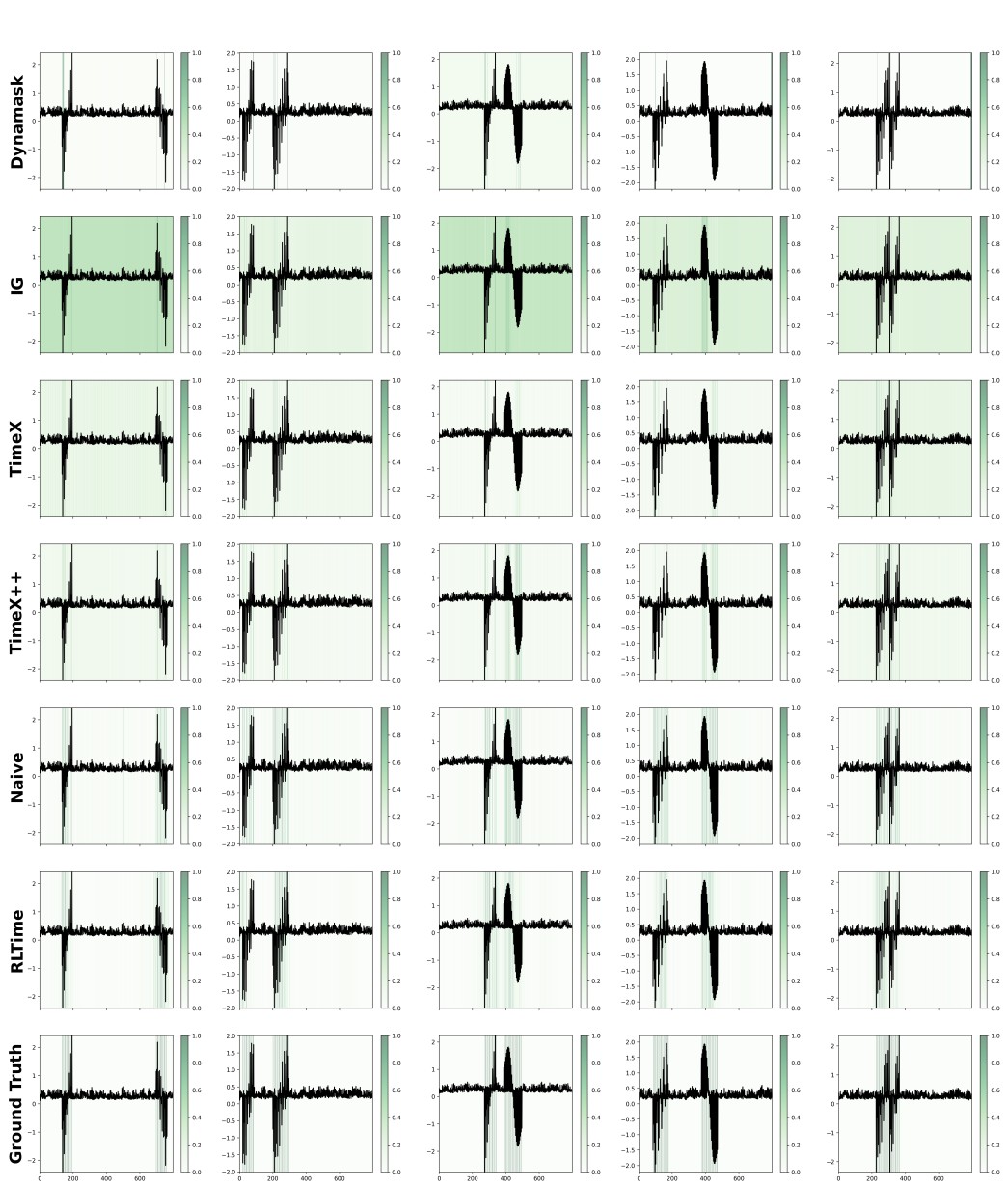

Figure 10: Visualization of all explainers on the SeqCombMV dataset. Each column corresponds to a unique and randomly selected sample. For each row, the method used to generate the corresponding explanation figure is indicated, with the ground truth explanations presented in the bottom row.

