# OpenReview forum: "RLTime: Reinforcement Learning-Based Feature Attribution for Interpretable Time Series Models"
_ICLR.cc/2026/Conference — Submitted to ICLR 2026_

### Official Review · Reviewer_c1g8 · 2025-10-29

**Soundness:** 3
**Presentation:** 2
**Contribution:** 3
**Rating:** 6
**Confidence:** 3

**Summary:**

This paper leverages reinforcement learning to study and explain the feature importance of time series data.

**Strengths:**

- This work leverages reinforcement learning to study and explain the feature importance of time series data, to bypass the limitations of continuous relaxation.
- Comprehensive experiments show the strong performance of the proposed method.

**Weaknesses:**

- Computational training: masked reconstruction model, reference model, and agent require training. Any one model without careful training would drop the performance.
- Figure 1 and Figure 2 lack clear explanations. It is good to provide key components and processes within the captions.
- No source code.

**Questions:**

- Is it possible to design the hybrid reward functions by combining input space and embedding space?
- Did you consider the out-of-distribution (OOD) issue for reconstruction model and reference model? If so, how did you handle it?
- You are focusing time series classification tasks in this work, is it possible to generalize to regression tasks?
- lines 83-85: what do you mean by "We first pretrain a masked reconstruction network that predicts the latent representation of a reference model."? I thought masked reconstruction network is only responsible for recovering time series, and has nothing to do with "predicts the latent representation".

---

> ### Author Response · Authors · 2025-11-24
> **Rebuttal**
>
> Hi Reviewer c1g8, Thanks for your detailed and constructive suggestions. In the following, we address your concerns separately.
>
> ### W1: Computational training
> Thanks for bringing this up. Accurately, RLTime is quite robust against a "not-optimal" setup, shown in the hyperparameter settings in Appendix B.7.
>
> ### W2: Better explanation in Figure 1 \& 2
> Thanks for mentioning this point. Below, we provide a better explanation of Figures 1 \& 2. We will update the following description as follows, with **bold** denoting the changed parts.
>
> `Figure 1: The RLTime training pipeline. **The agent starts with a fully masked state.** The agent sequentially selects features to unmask; unmasked features improve agreement with the reference model’s latent (blue). The agent is trained on a latent-shift reward (squared change between consecutive reconstructed latents), and its learned values yield feature importance.`
>
> `Figure 2: Diagram outlining our masked reconstruction training pipeline. **Here, the input $x$ and masked input $x \odot m$ are aligned through three losses in three spaces: input, latent, and logit, respectively.**
> `
> Please note that the PDF file has been revised accordingly.
>
> ### W3: Source code
> As stated in the *Reproducibility Statement*, the RLTime codebase is undergoing internal revisions and approval to ensure a robust and high-quality release. We commit to open-sourcing the source code within a reasonable time frame.
>
> ### Q1: hybrid reward design
> As we consider this, our primary goal is to ensure the model remains faithful to the reference model's interpretation of the data, rather than the raw data itself. Our concern is that incorporating input-space distances might inadvertently shift the focus toward information that the reference model doesn't heavily utilize. We feel the current approach best aligns with our core objective of explaining the model's understanding.
>
> ### Q2: OOD for reconstruction and reference models
> We appreciate this suggestion to explore out-of-distribution (OOD) settings. Our current work is situated within the established evaluation paradigm of in-distribution generalization, which allows for a direct comparison with prior research. We agree that OOD research is a critical and promising direction, and we see it as a natural next step for future work building upon the foundations laid here.
>
> ### Q3: Generalization to regression work?
> Thanks for this interesting idea. Our model proposes no constraints on classification or regression tasks. The only thing we need to alternate is to replace the prediction consistency loss $L_P$, which measures the distance in logit space, with other losses that deliver similar functionality on regression tasks, such as the MSE loss or Wasserstein distance.
>
> ### Q4: Clarification on lines 83-85
> We appreciate the opportunity to clarify this pipeline. The reconstruction model learns to recover full inputs from masked data (Eq. 1, Fig. 2), effectively learning what latent information can be restored. Its resulting reconstructed latent representations are then used to train the RL model (Eq. 3, Fig. 1). We will revise the text to articulate this flow more clearly.

---

### Official Review · Reviewer_2HDc · 2025-10-31

**Soundness:** 2
**Presentation:** 3
**Contribution:** 3
**Rating:** 4
**Confidence:** 3

**Summary:**

This paper (RLTime) presents a novel approach to time series feature attribution by formulating it as a sequential information acquisition task, effectively addressing the "soft-to-hard gap" in continuous relaxation interpretation methods. The proposed reward signal is based on local latent shifts in a pretrained reconstruction model, which is theoretically motivated and shows strong empirical performance. However, the framework introduces significant complexity through the two-stage training process, and the computational overhead during training is not reported or analyzed. Furthermore, the improved exploration appears to be trade-off against precision in several cases (Table 3).

**Strengths:**

1. RLTime works in the discrete action space (selecting features to reveal) during both training and evaluation. This design directly circumvents the "soft-to-hard gap" and "fractional leakage" issues associated with continuous relaxations (e.g., Gumbel-Softmax).

2. The use of reinforcement learning aims to learn a non-myopic acquisition policy that balances exploration and exploitation, potentially capture features that gradient-based methods might miss due to myopic exploration.

3. The reward is defined as the local latent shift , measuring the marginal impact of revealing a feature on the reconstructed latent representation of the reference model.Proposition.1 provides theorem by linking the expected latent shift to the expected drop in latent MSE (assuming a calibrated reconstructor).

4. The paper is well-motivated, well-structured and easy to follow.

**Weaknesses:**

1. (Major) Unreported Computational Overhead and Training Complexity:
The methodology requires a complex two-stage training process: pretraining the masked reconstruction network and then training the RL agent. The RL training phase involves generating trajectories and requires forward pass through the reconstruction model and reference encoder per-step to compute rewards, which seems to be computationally intensive, especially with a large action space ($T \times V$). While inference times are reported in Table 3, the paper did not cover any discussion of training times or a comparison of the overall computational cost against the baselines.

2. (Major) Gray-Box Requirement:
The entire framework depends on accessing the intermediate latent representation $z$ produced by the reference model's encoder. The framework also assumes the reference model can be clearly separated into an Encoder ($G$) and a Predictor ($F$) with a distinct latent bottleneck. It cannot be applied if the reference model’s internal activations are hidden, or if the architecture is monolithic.


3. (Major) Clarity of the Agent:
The MDP state is defined as $s_{t}=(x,m_{t})$, which includes the full input $x$. It is unclear what the RL agent actually observes. If the agent sees the full $x$, it is unclear how the mask $m_t$ effectively restricts the information available to the agent itself, rather than just restricting the input to the reconstruction model used for reward calculation. Also it is unclear if the policy is global across instances or trained per-instance?

4. (Minor) Performance Indicating Trade-offs:
Despite strong performance in recall (AUR), RLTime shows regressions in Area Under Precision (AUP) on synthetic datasets. It ranks last on SeqCombUV for AUP (0.7609 vs 0.9411 for the best baseline) in Table.1. This suggests that the improved exploration (higher AUR) achieved by the non-myopic RL approach may come at the cost of precision compared to the myopic baseline, a trade-off that is not discussed in the paper.

Clarifications are welcome and I will reconsider and raise the score if the questions are addressed.

**Questions:**

1. The method assumes the reference model has learned a well-structured and meaningful latent space. If the latent space is entangled or poorly regularized, will the explanations still faithfully reflect this internal state?

---

> ### Author Response · Authors · 2025-11-24
> **Rebuttal**
>
> Hi Reviewer 2HDc, thanks for your detailed response. Below, we address your concern and questions separately.
>
> ### W1: Computational Overhead and Training Complexity
> Thanks for mentioning this. We agree that training complexity is a crucial factor. Below, we conduct an ablation study against our two main baselines. We can observe that RLTime performs similarly to TimeX and TimeX++ in terms of runtime. We further add the following experimental results in Appendix B.6. In short, RLTime slightly increases the training complexity, but the overall training complexity is manageable.
>
> | Method | SeqCombUV | SeqCombMV | Epilepsy |
> | --- | --- | --- | --- |
> | TimeX | 232.9 | 469.5 | 619.5 |
> | TimeX++ | 312.3 | 454.2 | 594.3 |
> | RLTime-Rec | 321.2 | 412.2 | 522.1 |
> | RLTime-RL | 284.3 | 313.1 | 784.7 |
> | RLTime-total | 605.5 | 725.3 | 1306.8 |
>
> ### W2: Gray-Box Requirement
> Thank you for raising the point about the gray-box requirement. We appreciate the opportunity to clarify this aspect of our work. You are correct that RLTime operates under this assumption. We wish to note that this is consistent with the established baselines in our comparison [1,3], indicating that practitioners in this area are already familiar with this framework. To ensure our approach is robust and representative, we adopted the Transformer as a default and included an ablation study with CNN and LSTM models in Appendix B.4, following prior work [1-3]. These model types encompass the majority of current time-series classifiers and are designed to provide a latent space that serves as the standard operating environment for XAI methods in this field. Therefore, this requirement is both common and well-supported by the community.
>
> ### W3: Clarity of the Agent
> Thank you for highlighting this and providing us with the opportunity for further clarification. The RL policy's decision-making process is based on two key pieces of information: the original, unmasked input and the current state of the mask. This process is visualized in Figure 1. Additionally, we would like to clarify that the agent undergoes a single training phase for a given problem domain. Once trained, it operates as a global model, generalizing to new, unseen test instances without requiring any instance-specific training.
>
> ### W4: Performance Trade-offs
> Thank you for raising this important point about the AUP and AUR trade-off. We appreciate the opportunity to clarify. As you correctly noted, these metrics are indeed in a trade-off with each other. We discuss this around line 408 and provide the full analysis in Table 2 of the revised PDF. For this reason, we use the Area Under the Precision-Recall Curve (AUPRC) as our primary metric, as it integrates both precision and recall into a single, comprehensive measure of performance. This enables us to assess how effectively the model strikes this balance overall.
>
> ### Q1: What if the latent space is entangled
> Thank you for bringing this up. It is indeed an interesting question. We make no assumption about how the latent space is constructed. However, it is highly likely that some level of entanglement exists across the large set of datasets (7 datasets), reference model types (Transformers, CNNs, LSTMs), data splits (5 for each setting), and seeds (3 for each setting) we evaluate across. We observe that, in general, our approach consistently outperforms the baselines in all these settings, highlighting that our model is robust to the challenging settings commonly used to evaluate our XAI setting.
>
> ### Reference
> 1. [1] Owen Queen, et al. Encoding time-series explanations through self-supervised model behavior consistency, NeurIPS'23
> 2. [2] Zichuan Liu, et al. Timex++ learning time-series explanations with information bottleneck. ICML'24
> 3. [3] Jinghang Yue, et al. Optimal information retention for time-series explanations, ICML'25

---

### Official Review · Reviewer_Yzyd · 2025-10-31

**Soundness:** 3
**Presentation:** 3
**Contribution:** 2
**Rating:** 4
**Confidence:** 3

**Summary:**

This paper studies interpretable time series models and highlights two key limitations of existing approaches, namely myopic exploration and the soft-to-hard gap. To address these limitations, the authors present RLTime, a framework that formulates attribution as a sequential information acquisition task and directly operates in a discrete action space. The framework integrates a masked reconstruction network and a distributional reinforcement learning agent with a well-defined reward. Finally, RLTime is evaluated on both synthetic and real-world datasets, demonstrating improved interpretability and predictive performance.

**Strengths:**

**S1.** The paper addresses a practically important problem, aiming to improve explainability in deep time series modeling while maintaining predictive accuracy. Enhancing transparency in such models is valuable for building trust and supporting accountable decision processes.

**S2.** The RLTime framework re-formulates attribution as sequential information acquisition and uses a reinforcement learning agent with a clearly defined reward, a masked reconstruction network, and a reference model encoder. From a technical perspective, the design is coherent and the components are integrated in a logical manner.

**S3.** The experimental evaluation compares RLTime with representative baselines on both synthetic and real-world datasets. The study includes occlusion analyses, search space investigation, ablations, and visualization experiments, providing a relatively comprehensive empirical assessment.

**Weaknesses:**

**W1.** While the paper discusses the two challenges in existing models, namely myopic exploration and the soft-to-hard gap, the motivation could be further strengthened. In particular, including empirical observations or case studies that demonstrate these issues in practice would help clarify their impact and better motivate the core design of the proposed framework.

**W2.** In the experimental evaluation, it would be helpful to analyze the “failure cases” in greater depth. For instance, RLTime shows lower performance than TimeX++ under small thresholds in Figures 4 (b), (f), and (h). In addition, RLTime does not surpass the baseline models on the Boiler dataset in Table 2. A discussion of these cases, including possible reasons and insights, would strengthen the understanding of the framework’s behavior and limitations.

**W3.** In Table 3, RLTime appears to consistently underperform Naïve Search on the AUP metric. This raises the question of whether different methods are making different trade-offs between AUP and AUR, with RLTime potentially favoring AUR at the cost of AUP. It would be useful to clarify which metrics are considered primary for model comparison, and to provide further discussion on this trade-off to help readers interpret the results.

**W4.** The dataset selection across different experiments in both the main paper and the appendices appears inconsistent. Without a clear justification, this variation makes it more difficult to interpret the results in a unified manner and may affect the perceived coherence of the evaluation. To strengthen the credibility of the experimental findings, it would be helpful either to adopt consistent dataset choices across analyses or to clearly explain the rationale behind the selection for each experiment.

**W5.** It would be beneficial to include an analysis of key hyperparameters to demonstrate the robustness of RLTime under different settings. Such results would provide a clearer picture of the framework’s sensitivity and stability.

**Questions:**

Beyond W1-W5, I have the following questions for clarification:

**Q1.** In Section 3.3, the paper explains the benefits of the proposed local difference formulation when defining the reward, compared to a global formulation. It would be helpful to present a comparison with the global alternative, i.e., the residual gap to the fully unmasked embedding, to provide empirical evidence supporting this design choice.

**Q2.** In Section 3.4, the initial state assumes that all entries are masked. Further explanation on how training is initiated from this state would improve clarity, especially regarding how the reinforcement learning agent begins meaningful exploration under full masking.

**Q3.** In the discussion of explainable reinforcement learning in Section 5, the connection between RLTime and existing methods in this category should be made more explicit. This would help position RLTime within the broader literature and clarify its conceptual differences or similarities.

**Q4.** Appendix C suggests that RLTime and Naïve Search are both proposed in this work, with RLTime being more explorative and Naïve Search being more exploitative. If this interpretation is correct, a clearer explanation of their relationship would be valuable. In addition, practical guidance on when each approach would be preferable would support adoption by practitioners.

---

> ### Author Response · Authors · 2025-11-24
> **Rebuttal**
>
> Hi Reviewer Yzyd, thanks for your detailed response. In the following, we address your concerns and questions separately.
>
> ### W1: Include empirical observations or case studies in the introduction.
> Thank you for engaging with our intuition. To provide more concrete support, we have included visual examples in Appendix C that we believe offer an intuitive illustration of both points.
>
> - Regarding myopic exploration, prior work [1, 2, 3] often shows that baselines select only a subset of the ground truth features, a tendency that is also visible in the results from [3]. In our experiments, RLTime appears to achieve a higher overlap with the ground truth, suggesting a more comprehensive exploration.
> - For the soft-to-hard gap, we note that baseline methods often assign low importance values to many positions. In contrast, our method tends to produce more decisive attributions.
>
> We have revised the introduction and appendix to clarify these points, and we hope that these changes, along with the visualizations, make our rationale easier to follow.
>
>
> ### W2: Further explanation on occlusion experiments
> Thank you for this insightful observation. This is a known challenge in the field, as similar cases can be found in previous work (e.g., Figure 3 in [1]) due to dataset complexity. We have added a discussion on this very point in the revised manuscript to acknowledge and address it.
>
> ### W3: Trade-off between AUP and AUR & Q4: A clearer explanation between RLTime and Naive Search
> Thanks for the detailed comparison. The primary solution is RLTime, as determined by its better performance on AUPRC, which represents how well a model balances precision (correctness of positive predictions) and recall (coverage of true positives) by integrating precision at every level of recall. As for the specific metrics, namely AUP and AUR, it is hard to determine which is more important, as it tends to be domain-dependent. For instance, in attributing to medical or system time series, recall tends to be favoured.
>
> ### W4: Clarification on dataset selection
> We appreciate the opportunity to clarify our dataset selection. The main paper uses established benchmarks from [1-3] for comparability, while the appendix uses a consistent mix of synthetic and real-world datasets (Appendix B) to probe specific model components. We will further clarify this in the revised version.
>
> ### W5: Key hyperparameters analysis
> Thanks for mentioning this key aspect of RLTime. We append an ablation study in Appendix B.7 of the revised manuscript. We aim to extend this study in a future version. In short, RLTime proves to be robust against "non-optimal" hyperparameters.
>
> ### Q1: Difference between local \& global reward
> Thanks for bringing this up. We conducted an ablation study (Appendix B.2) investigating the reward design, with formats 2 and 3 representing different global reward designs. In short, local reward designs tend to outperform global reward designs. This is likely because once the RL agent reaches a "good" state, the local reward only measures the next action, while the global reward continues to generate high rewards even if the RL agent wanders around the "good" state.
>
> ### Q2: How is the initial stage unmasked?
> This initial unmasking follows the same principle as all the RL agent's decisions: it directly chooses a position in $x$ to unmask based on its prediction of future rewards.
>
> ### Q3: Further clarification between RLTime and existing methods
> Thanks for pointing out this issue. The sub-title "Explainable Reinforcement Learning" in Section 5 could be easily misunderstood. This subsection is intended to discuss previous works that utilize RL for explainability, where RLTime belongs. We have revised the subsection with the differences listed below:
>
> ```
> \paragraph{Reinforcement Learning for Explainability} Reinforcement learning has been widely utilized to provide explanations in various domains. INVASE ...
> ```
>
> ### Reference
> 1. [1] Owen Queen, et al. Encoding time-series explanations through self-supervised model behavior consistency, NeurIPS'23
> 2. [2] Zichuan Liu, et al. Timex++ learning time-series explanations with information bottleneck. ICML'24
> 3. [3] Jinghang Yue, et al. Optimal information retention for time-series explanations, ICML'25

---

### Meta-Review · Area_Chair_hKA4 · 2025-12-31

**Summary:**

In this paper, we present RLTime, a reinforcement learning-based framework for time series feature attribution. The essential novelty of the method is tackling two general drawbacks of existing approaches: myopic exploration and the soft-to-hard gap in continuous relaxations. RLTime employs a masked reconstruction network integrated with a distributional reinforcement learning agent to sequentially acquire information on feature importance. The authors assessed RLTime performance on synthetic and real-world datasets, reporting improved interpretability and predictive performance.

**Reviewer Concerns:**

The reviewers (Reviewer 2HDc, Reviewer c1g8) were worried about the intricacies of RLTime’s two-stage training process, as it pretreats the masked reconstruction network before advancing to the RL agent. Reviewer 2HDc challenged the gray-box need and argued that RLTime is assuming access to the intermediate latent representation of the reference model. It was not clear how the reinforcement learning agent operates in the state space, nor was it clear if the mask effectively narrows the information available to the agent. As several reviewers (Reviewer 2HDc, Reviewer Yzyd) point out, RLTime also has a trade-off between Area Under Precision (AUP) and Area Under Recall (AUR), i.e., RLTime did worse in some cases on AUP. Reviewer Yzyd had noticed a lack of consistency between the dataset selection of the main paper and the appendices. The authors made a passing reference to where RLTime belongs in the wider exploratory literature on explainable reinforcement learning, but failed to fully connect RLTime to methods out there.

**Reviewer Scores:**

The final scoring would remain unchanged as reviewed by Yzyd since the issue with inconsistent datasets, computational complexity, and the performance trade-off.
Reviewer 2HDc's score would likely have remained 4 even after the rebuttal. The complexity and gray-box requirement remain significant issues, and the lack of a detailed discussion on the trade-offs between AUP and AUR or the robustness of RLTime's performance prevents the score from improving. There was also no new evidence presented to address the computational overhead concerns in detail.
Reviewer c1g8 had rated the paper positively at first. However, the authors did not respond to some of the weaknesses. For example, there was no responses provided for “Any one model without careful training would drop the performance” from Reviewer c1g8.

---

### Decision · Program_Chairs · 2026-01-26

Reject